# Robust Ego-Exo Correspondence with Long-Term Memory

**Yijun Hu**[1*]    **Bing Fan**[2*]    **Xin Gu**[1]    **Haiqing Ren**[3,1]    **Dongfang Liu**[4]
**Heng Fan**[2†]    **Libo Zhang**[3,1†]
[1]University of Chinese Academy of Sciences    [2]University of North Texas
[3]Institute of Software Chinese Academy of Sciences    [4]Rochester Institute of Technology
[*]Equal contribution and co-first authors    [†]Equal advising and corresponding authors

## Abstract

Establishing object-level correspondence between egocentric and exocentric views is essential for intelligent assistants to deliver precise and intuitive visual guidance. However, this task faces numerous challenges, including extreme viewpoint variations, occlusions, and the presence of small objects. Existing approaches usually borrow solutions from video object segmentation models, but still suffer from the aforementioned challenges. Recently, the Segment Anything Model 2 (SAM 2) has shown strong generalization capabilities and excellent performance in video object segmentation. Yet, when simply applied to the ego-exo correspondence (EEC) task, SAM 2 encounters severe difficulties due to ineffective ego-exo feature fusion and limited long-term memory capacity, especially for long videos. Addressing these problems, we propose a novel EEC framework based on SAM 2 with long-term memories by presenting a dual-memory architecture and an adaptive feature routing module inspired by Mixture-of-Experts (MoE). Compared to SAM 2, our approach features **(i)** a Memory-View MoE module which consists of a dual-branch routing mechanism to adaptively assign contribution weights to each expert feature along both channel and spatial dimensions, and **(ii)** a dual-memory bank system with a simple yet effective compression strategy to retain critical long-term information while eliminating redundancy. In the extensive experiments on the challenging EgoExo4D benchmark, our method, dubbed ***LM-EEC***, achieves new state-of-the-art results and significantly outperforms existing methods and the SAM 2 baseline, showcasing its strong generalization across diverse scenarios. Our code and model are available at `https://github.com/juneyeeHu/LM-EEC`.

## 1 Introduction

Aligning observations of objects in egocentric and exocentric viewpoints can facilitate many applications. In augmented reality (AR), a person wearing smart glasses could quickly pick up new skills with a virtual intelligent coach that provides real-time guidance. In robotics, a robot watching people in its environment is able to acquire new dexterous manipulation skills with less physical experience. The recent work EgoExo4D [1] makes significant contributions to establishing object-level correspondence between these ego-exo views by introducing a benchmark featuring annotated, temporally synchronized egocentric and exocentric videos along with object segmentation masks.

The first-person (ego) perspective captures fine-grained details of hand-object interactions and the camera wearer's attention, while the third-person (exo) perspective provides a broader view of full-body poses and the surrounding environment. Therefore, this task presents significant challenges, including extreme viewpoint variations, substantial object occlusions, and the presence of numerous small objects. Previous approaches, such as XSegTx [1], which formulates the task as a matching

39th Conference on Neural Information Processing Systems (NeurIPS 2025).

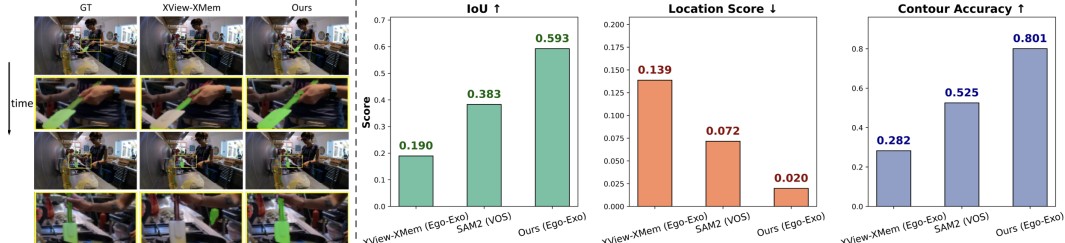

Figure 1: **Left:** Comparison of segmentation results between XView-XMem and our model, using exocentric videos as an example. **Right:** Quantitative results on the EgoExo4D validation set.

problem, and XView-XMem [1], which adapts XMem [2] to track objects across different views, struggle to effectively address these challenges as depicted on the left of Figure 1, highlighting the need for more robust solutions. Recently, a unified model for both video and image segmentation, called Segment Anything Model 2 (SAM 2) [3], has been introduced. Trained on the SA-V dataset [3], which contains 50.9K videos with 642.6K masklets, SAM 2 demonstrates strong generalization capabilities and adaptability across a wide range of scenarios and tasks [4, 5]. The right panel of Figure 1 shows the strong performance of SAM 2 on our task. "Ego-Exo" refers to segmenting objects in the exocentric videos, given the corresponding egocentric videos and annotations, while "VOS" refers to segmenting objects in the exocentric videos using only the first-frame annotations.

However, SAM 2 still faces challenges in establishing object-level correspondence between temporally synchronized egocentric and exocentric videos. A key issue arises when using annotations from one view as prompts to segment the other. Effective fusion between two views is crucial, yet SAM 2 simply adds memory-aware features and prompt embeddings together, completely overlooking the gap in different features and distribution differences between two views. Another major challenge is the scale of the task, which involves processing large volumes of long videos. Yet SAM 2 retains only a few frames closest to the current frame in its memory bank. As a result, long-term information is not effectively preserved, leading to degradation in complicated scenarios.

To overcome these limitations, we first introduce an adaptive Mixture-of-Experts (MoE) strategy that dynamically integrates diverse knowledge-fused features based on their distinct characteristics, enabling them to complement each other effectively. Additionally, optimizing SAM 2's memory management is crucial. Current approaches [3, 5, 6, 7], which directly store recent or selected frames in the memory bank, introduce redundant features while failing to retain long-term information. Addressing these challenges is essential to fully harness SAM 2's capabilities for robust object segmentation in ego-exo correspondence scenarios.

we propose LM-EEC, a SAM-based robust segmentation model with dual compressed long-term memory and an adaptive fusion module. Our method introduces two key advancements: **(i)** a flexible Memory-View Mixture-of-Experts module that dynamically reweights expert features based on their characteristics, enabling more effective integration of complementary information, and **(ii)** a dual-memory bank system that separates ego and exo memory banks and leverages a specially designed compression strategy to efficiently preserve critical long-term information. In our extensive evaluation on the challenging EgoExo4D benchmark, LM-EEC achieves new state-of-the-art results and significantly outperforms other models including SAM 2.

In summary, our main *contributions* are as follows: ♠ We enhance SAM 2's segmentation capabilities for the ego-exo correspondence task by introducing a specialized MoE mechanism that effectively integrates features from both egocentric and exocentric views; ♥ To efficiently manage long-term dependencies, we design a dual-memory bank system that incorporates a view-specific compression strategy, effectively reducing redundancy while retaining critical long-term information; ♣ Extensive experiments on EgoExo4D demonstrate that our model achieves state-of-the-art performance, showcasing its strong generalization ability across diverse real-world scenarios.

## 2   Related work

**Video object segmentation**. Video Object Segmentation (VOS) involves tracking an object throughout a video given its mask in the first frame [8]. This task is classified as "semi-supervised VOS"

since the initial object mask serves as a supervision signal available only in the first frame. VOS has attracted significant attention due to its broad applications in areas such as video editing and robotics.

Early deep learning-based models often rely on online fine-tuning [9, 10, 11, 12, 13, 14, 15, 16], either on the first frame or across all frames, to adapt the model to the target object. To improve inference speed, offline-trained models were introduced, leveraging conditioning on either the first frame alone [17, 18] or incorporating information from previous frames [19, 20]. These conditioning strategies have since been extended to all frames using RNNs [21] and transformers [22, 23, 24, 25, 2, 26, 27].

Recently, the Segment Anything Model 2 (SAM 2) [3] has emerged as a unified foundational model for promptable object segmentation in both images and videos. Notably, SAM 2 [3] has established new state-of-the-art benchmarks across various VOS tasks, significantly outperforming previous methods. For instance, SAM2-UNet [4] introduces a simple yet effective U-shaped architecture for versatile segmentation across both natural and medical domains, while SAMURAI [5], an enhanced adaptation of SAM 2 tailored for visual object tracking, achieves substantial improvements in both success rate and precision over existing tracking models. Building upon this foundation, we develop our model based on SAM 2 [3] to further enhance segmentation performance in our task.

**Long-term video models**. Long-term video understanding focuses on capturing long-range dependencies in the videos. The primary challenge lies in balancing the retention of crucial information while maintaining computational efficiency. To address this, a common strategy involves utilizing pre-extracted features, eliminating the need for joint training of backbone architectures [28, 29, 30, 31, 32]. Alternatively, some studies have explored sparse video sampling techniques [33, 34], which reduce the number of input frames by selectively preserving salient content.

Another widely adopted approach is the use of memory banks to retain relevant features, particularly in video object segmentation. For instance, XMem [2] employs multiple independent yet highly interconnected feature memory stores: a rapidly updated sensory memory, a high-resolution working memory, and a compact but persistent long-term memory. Similarly, RMem [35] and SAM 2 [3] constrain memory banks to a limited set of essential frames, striking a balance between relevance and freshness when updating feature representations.

## 3 The Proposed Method

### 3.1 Task definition

Given a pair of synchronized ego-exo videos and a sequence of query masks for an object of interest in one of the videos, the objective is to identify the corresponding masks of the same object in each synchronized frame of the other view, if visible. This task consists of two scenarios: one where the ego view serves as the query to segment the object in the exo view, and the other where the exo view serves as the query to segment the object in the ego view. These two settings are referred to as *Ego-to-Exo correspondence* and *Exo-to-Ego correspondence*, denoted as *Ego2Exo* and *Exo2Ego*, respectively, in the following for brevity.

Formally, taking *Ego2Exo* as an example, let an ego-view video with $T$ frames and a corresponding exo-view video with $T$ frames be represented as $\{I_t^{ego}, G_t^{ego}\}_{t=1}^T$ and $\{I_t^{exo}\}_{t=1}^T$, respectively. Here, $G_t^{ego}$ denotes the ground-truth object mask of frame $I_t^{ego}$. Given these synchronized frames, the goal is to segment the object masks $M_t^{exo}$ in each frame of the exo-view video.

### 3.2 Preliminary on SAM 2

SAM 2 [3] begins with an image encoder that encodes each input frame into embeddings. In contrast to SAM [36], where frame embeddings are directly fed into the mask decoder, SAM 2 designs a memory module that conditions features of the current frame on both previous and prompted frames.

Specifically, for semi-supervised video object segmentation tasks, SAM 2 maintains a memory bank at each time step $t \geq 1$:

$$\mathcal{M}_t = \left\{ \mathbf{M}_\tau \in \mathbb{R}^{K \times C} \right\}_{\tau \in \mathcal{I}} \tag{1}$$

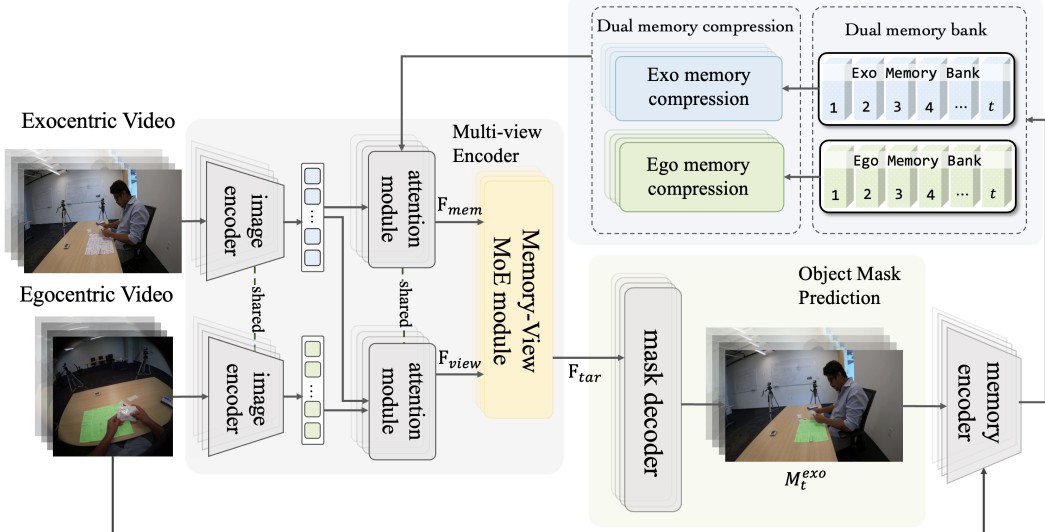

Figure 2: Overview of our proposed model, which consists of three key components: multi-view encoding, dual memory compression, and object mask prediction. The multi-view encoder extracts features from egocentric and exocentric videos, using a Memory-View MoE module to adaptively combine memory-aware and view-specific representations. The object mask prediction module then generates segmentation masks for objects in the exocentric view. To capture long-term dependencies efficiently, we apply dual memory compression on the memory banks from both viewpoints.

where K is the number of memory tokens per frame, C is the channel dimension, and $\mathcal{I}$ is the set of frame indices included in the memory. In SAM 2, the memory bank stores up to N of the most recent frames, along with the initial mask, using a First-In-First-Out (FIFO) queue mechanism.

The encoded frame feature would be fused with the memory bank through the memory attention module, which is stacked by L transformer blocks, facilitating full interaction between the two. After this memory fusion, if mask prompts are present, the fused feature is summed with the dense prompt embedding, which is then processed by the convolutions of the prompt encoder.

## 3.3 Overview

Figure 2 provides an overview of our proposed model LM-EEC. Taking ***Ego2Exo*** as an example, given the frames and target object masks from the ego view along with the corresponding frames from the exo view (left side of Figure 2), the workflow then unfolds as follows:

First, the video frames from both views are encoded using the image encoder. The feature representations of the current exo and ego frames are then passed through the dual attention module, where the exo feature is integrated separately by the attention module with the stored features from both the ego and exo memory banks, as well as the corresponding annotated ego feature which is encoded by the off-the-shelf memory encoder. Once the memory-aware and view-specific features are obtained, the Memory-View Mixture-of-Experts module(sec 3.4) adaptively integrates the two expert features. The fused feature is subsequently processed by the mask decoder to generate the predicted object mask.

Finally, the predicted mask, along with the given ego-view mask and their corresponding encoded features, are fed into the memory encoder to generate memory frames, which are stored in the ego and exo memory banks. As the dual memory banks have a fixed capacity, the tailored memory compression mechanism(sec 3.5) is triggered when the stored frames exceed this limit, ensuring efficient memory management.

## 3.4 Memory-View Mixture-of-Experts module

SAM 2 simply adds together the prompt embedding generated by the prompt encoder and the memory-aware feature. In our task, the memory-aware feature and the view-specific feature, which

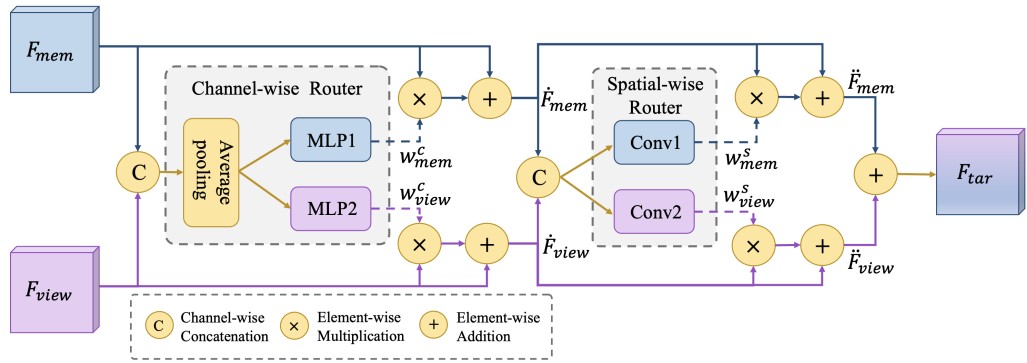

Figure 3: Overview of the proposed Memory-View Mixture-of-Experts (MV-MoE) module. Channel- and spatial-wise routers generate dynamic weights to recalibrate memory-aware and view-specific features, enabling adaptive and context-aware fusion of complementary information.

serves as prompt information, highlight different regions and have different feature distributions. This may cause the prompt information to overwhelm the representation of the other branch, leading to less discriminative and robust target representations.

To alleviate this issue, we propose the **Memory-View Mixture-of-Experts (MV-MoE) module**, which treats the memory-aware and view-specific features as two complementary experts. Rather than performing direct fusion, we design a lightweight dual-branch routing mechanism to adaptively assign contribution weights to each expert along both channel and spatial dimensions, allowing the network to determine the relative importance of each expert in a data-dependent manner.

Specifically, as shown in Figure 3, given the memory-aware feature $F_{mem} \in \mathbb{R}^{h \times w \times c}$ and view-specific feature $F_{view} \in \mathbb{R}^{h \times w \times c}$, channel-wise routing is performed firstly. We concatenate the two features along the channel dimension and apply global average pooling to obtain a compact descriptor. This descriptor is then passed through two separate MLPs—each consisting of two linear layers, with ReLU and sigmoid activations—to generate dynamic channel-wise importance weights $\mathbf{w}_{mem}^c, \mathbf{w}_{view}^c \in \mathbb{R}^{1 \times 1 \times c}$. These weights are used to recalibrate each expert through residual modulation:

$$\mathbf{w}_{mem/view}^c = MLP_{1/2}(Avg(Concat(F_{mem}, F_{view}))), \qquad (2)$$

$$\dot{F}_{mem/view} = \mathbf{w}_{mem/view}^c \otimes F_{mem/view} + F_{mem/view}, \qquad (3)$$

where $Concat(\cdot)$ and $Avg(\cdot)$ denote channel-wise concatenation and global average pooling, respectively, and $\otimes$ represents element-wise multiplication. This operation adaptively emphasizes the more informative channels in each expert while preserving the original feature content.

Subsequently, spatial-wise routing is applied to the channel-enhanced features $\dot{F}_{mem}$ and $\dot{F}_{view}$. The two features are concatenated and fed into two parallel convolutional branches—each composed of a Conv–ReLU–Conv–Sigmoid sequence—to generate spatial attention maps $\mathbf{w}_{mem}^s, \mathbf{w}_{view}^s \in \mathbb{R}^{h \times w \times 1}$. These spatial weights are then used to further refine each feature spatially, again using residual modulation:

$$\mathbf{w}_{mem/view}^s = Conv_{1/2}(Concat(\dot{F}_{mem}, \dot{F}_{view})), \qquad (4)$$

$$\ddot{F}_{mem/view} = \mathbf{w}_{mem/view}^c \otimes \dot{F}_{mem/view} + \dot{F}_{mem/view}, \qquad (5)$$

Finally, the refined memory-aware and view-specific features are summed to produce the fused target feature which is then forwarded to the decoder for subsequent segmentation:

$$F_{tar} = \ddot{F}_{mem} + \ddot{F}_{view}. \qquad (6)$$

These routing strategies enable the model to adaptively modulate the contributions of memory-aware and view-specific features across both spatial and channel dimensions, aligning with the core intuition of the Mixture-of-Experts (MoE) paradigm—selectively leveraging complementary sources of knowledge based on the input context. Unlike conventional MoE architectures that rely on multiple sub-networks and sparse expert activation, our design performs **dense, feature-level expert routing**, which preserves the adaptive expert weighting behavior of MoE while avoiding the complexity of network-level sparsity. This fine-grained and lightweight formulation enables context-aware fusion and enhances the model's flexibility in dynamic video scenarios.

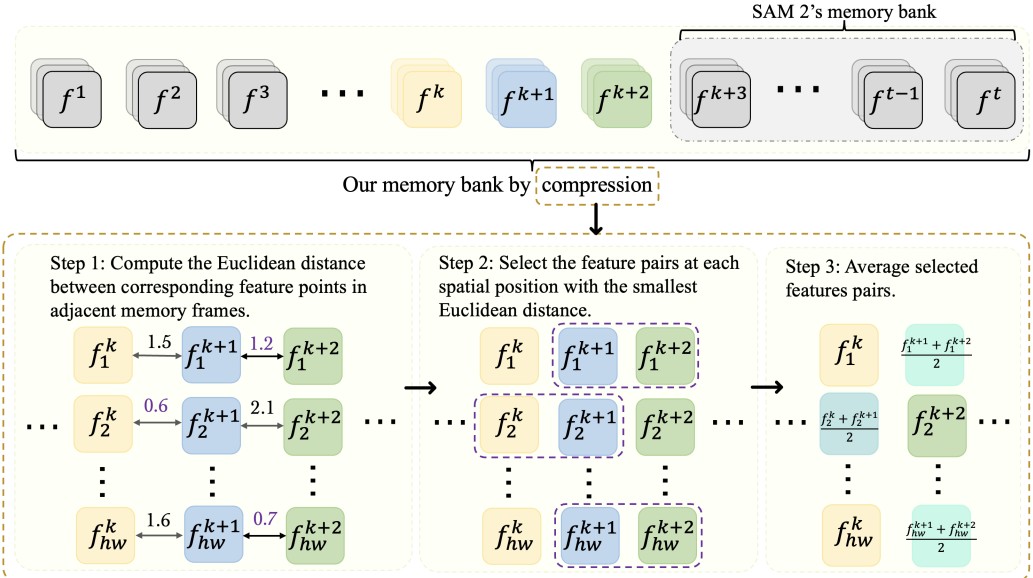

Figure 4: An illustration of our memory bank compression strategy, which preserves a fixed memory size in both ego-view and exo-view memory banks by aggregating temporally redundant information.

## 3.5 Dual Memory Compression

Previous work [1] constructs a single memory bank that stores both egocentric and exocentric features uniformly. However, egocentric and exocentric videos differ substantially in terms of viewpoints, motion patterns, and visual appearance. To better leverage their complementary characteristics, we first design a dual memory bank based on SAM 2, in which ego and exo features are stored separately.

SAM 2 maintains the most recent N frame features in its memory bank. Although effective in the short-term context, this strategy suffers from several limitations. As segmentation progresses, the memory bank gradually loses alignment with earlier frames, reducing its temporal consistency. Meanwhile, since the target object typically occupies only a small region of each frame, the memory bank becomes saturated with redundant spatial information. More critically, this unified memory design fails to differentiate between egocentric and exocentric views, thereby undermining our dual-memory objective.

To address these challenges, we propose a view-specific memory compression strategy that separately compresses the ego and exo memory banks. This approach reduces redundancy, preserves discriminative long-term features, and improves both efficiency and robustness.

Figure 4 illustrates our memory compression process. The core idea is to temporally aggregate and compress video features by exploiting similarities between adjacent frames, while retaining informative long-term content. This enables us to maintain compact and discriminative memory representations for both ego and exo views.

The compression algorithm is applied at each auto-regressive iteration whenever the length of the ego/exo memory bank exceeds a pre-defined threshold $M$. Formally, given an ego/exo memory bank consisting of the following sequence:

$$[f^1, f^2, \dots, f^M], \quad f^t \in \mathbb{R}^{P \times C}, \tag{7}$$

where $P = h \times w$, representing the spatial dimensions of each memory frame feature. When a new frame feature $f^{M+1}$ arrives, the memory bank must be compressed by reducing its length by one.

For each spatial location $i$, we first compute the Euclidean distance between the corresponding feature points in the temporally adjacent memory frames:

$$d_i^t = \text{Eucli}(f_i^t, f_i^{t+1}), \quad t \in [1, M], \quad i \in [1, P], \tag{8}$$

Next, we identify the most temporally redundant features by selecting the feature with the minimum Euclidean distance, indicating the highest similarity across time:

$$k = \arg\min(d_i^t), \quad t \in [1, M]. \tag{9}$$

Finally, to reduce the memory bank length by one, we perform a simple feature averaging at each spatial location:

$$f_i^k = \frac{f_i^k + f_i^{k+1}}{2}. \tag{10}$$

This three-in-one approach not only reduces redundancy but also preserves long-term information. Furthermore, it adaptively updates the memory bank based on scene variations in each viewpoint, ensuring a compact and informative memory representation.

## 4 Experiments

### 4.1 Dataset

We conduct experiments on the challenging EgoExo4D benchmark [1]. EgoExo4D is a dataset specifically designed for the ego-exo correspondence task. It contains approximately 5.5K annotated objects across 1.3K temporally synchronized egocentric and exocentric video pairs. In total, around 4 million frames have been annotated, resulting in 742K ego-view and 1.1M exo-view paired segmentation masks. We adopt the official dataset split in our experiment, where 756 videos are used for training, 202 for validation, and 291 for testing.

### 4.2 Metrics

Following [1], we employ four evaluation metrics, including Intersection Over Union (**IoU**) between the predicted and ground-truth masks, Location Error (**LE**), which is defined as the normalized distance between the centroids of the predicted and ground-truth masks, Contour Accuracy (**CA**), which measures how well the predicted masks match the ground-truth masks on the boundary, and Existence Balance Accuracy (**BA**), which evaluates the method's ability to estimate object visibility in the target view, since, in practice, objects may often be occluded or fall outside the field of view.

### 4.3 Implementation Details

Our model is built upon the official SAM 2 base [3], with key modifications including the introduction of Memory-View MoE module and a refined memory bank construction strategy. Following the training protocol of SAM 2, we sample 8 consecutive frames for each object from ego-exo video pairs, and set the memory bank size to 6. To reduce computational overhead due to the large resolution of original video frames, we resize all frames to $480 \times 480$, following the practice in [1]. Given the large-scale nature of EgoExo4D, we train all modules jointly based on the pre-trained SAM 2 checkpoint, without freezing any components. The model is trained for 60 epochs on 8 NVIDIA A100 GPUs with a batch size of 32 and evaluated on a single V100 GPU, achieving an inference speed of approximately 8.4 FPS. Memory compression is applied only during inference. Our code and model are available at `https://github.com/juneyeeHu/LM-EEC`.

### 4.4 Comparison to State-of-the-art Methods

**Table 1** presents the quantitative results in comparison with previous methods on the **test set**. We experiment with two settings: providing the ground-truth object track in the exo view (exo query mask) and predicting it in the ego view **Exo2Ego**), and vice versa. Given the limited availability of existing models tailored for this task, we selected several VOS models from 2021 onward [37, 38, 39, 40, 41] and adjusted their architectures in a manner similar to XView-XMem [1], namely, integrating features from the other view with annotations before segmenting the current frame. All models are trained or fine-tuned on the EgoExo4D dataset to ensure a fair comparison. To provide a more comprehensive comparison, we additionally report the performance of our **base model**, which comprises only a standard dual-memory bank and simple prompt addition, to intuitively highlight the effectiveness of our proposed method. As shown in Figure 5, we also evaluate and compare the segmentation

Table 1: Comparison for the ego-exo correspondence benchmark on **test set**. Best results are reported in **bold**, whereas our results are highlighted in `cyan` and the results of the base model are underlined.

| Query Mask | Method | IoU↑ | LE↓ | CA↑ | BA↑ |
|---|---|---|---|---|---|
| Ego | XSegTx [1] | 18.99 | 0.070 | 0.386 | 66.31 |
| Ego | XView-Xmem (w/ finetuning) [1] | 14.84 | 0.115 | 0.242 | 61.24 |
| Ego | XView-Xmem (+ XSegTx) [1] | 34.90 | 0.038 | 0.559 | **66.79** |
| Ego | STCN [37] | 27.39 | 0.109 | 0.378 | 61.97 |
| Ego | QDMN [38] | 27.03 | 0.108 | 0.346 | 63.16 |
| Ego | GSFM [39] | 3.98 | 0.146 | 0.057 | 52.39 |
| Ego | SimVOS [40] | 38.26 | 0.090 | 0.481 | 57.21 |
| Ego | Cutie [41] | 27.03 | 0.108 | 0.346 | 59.18 |
| Ego | Base model | 52.13 | 0.024 | 0.734 | 61.56 |
| Ego | LM-EEC(Ours) | **54.98** | **0.017** | **0.778** | 64.22 |
| Exo | XSegTx [1] | 27.14 | 0.104 | 0.358 | **82.01** |
| Exo | XView-Xmem (w/ finetuning) [1] | 21.37 | 0.139 | 0.269 | 61.72 |
| Exo | XView-Xmem (+ XSegTx) [1] | 25.00 | 0.117 | 0.327 | 59.71 |
| Exo | STCN [37] | 27.61 | 0.122 | 0.348 | 65.98 |
| Exo | QDMN [38] | 17.56 | 0.158 | 0.213 | 58.94 |
| Exo | GSFM [39] | 13.78 | 0.164 | 0.169 | 59.53 |
| Exo | SimVOS [40] | 40.67 | 0.099 | 0.481 | 66.62 |
| Exo | Cutie [41] | 47.52 | 0.070 | 0.579 | 70.71 |
| Exo | Base model | 57.27 | 0.047 | 0.677 | 57.11 |
| Exo | LM-EEC(Ours) | **65.77** | **0.031** | **0.774** | 58.14 |

Figure 5: Performance evaluation across different object sizes in the target (exo) view, including IoU, shape accuracy, and location score.

performance of our method across varying object sizes under the ***Ego2Exo*** task. Specifically, we divide the **test set** based on the proportion of object pixels within each frame. Furthermore, we provide visual results, conduct qualitative comparisons with other baselines, and analyze the model's performance across diverse activity scenarios in the EgoExo4D dataset. For more details, please kindly refer to the ***appendix***.

Overall, our model consistently outperforms all baselines across different object sizes and delivers substantial improvements over the base model, particularly in the ***Exo2Ego*** task. XSegTx [1] achieves a high level of balanced accuracy because it is a co-segmentation model and employs specialized data augmentation techniques during training.

### 4.5 Ablations

We conduct ablation studies on the **validation set** of EgoExo4D for the ***Ego2Exo*** task. Evaluation metrics are reported, with our final configuration highlighted in `cyan`.

**MV-MoE module** Table 2 presents the ablation study on our MV-MoE module. We compare two different experimental settings: (1) excluding the prompt from the other view of the current frame, just using the memory-aware feature for segmentation, (2) directly adding the prompt and the memory-aware feature, as done in SAM 2. Our results indicate that simple prompt addition fails to fully exploit the effectiveness of the prompts. In contrast, our proposed module leverages a routing

Table 2: Ablation on our fusion mechanism.

| Setting | IoU↑ | LE↓ | CA↑ |
|---|---|---|---|
| w/o other view | 0.5691 | 0.0276 | 0.7660 |
| simply add (base model) | 0.5673 | 0.0258 | 0.7643 |
| MV-MoE | **0.5925** | **0.0198** | **0.8006** |

Table 3: Ablation on training frame number.

| Frame | IoU↑ | LE↓ | CA↑ |
|---|---|---|---|
| 4 | 0.5857 | 0.0204 | 0.7931 |
| 8 | 0.5925 | 0.0198 | 0.8006 |
| 10 | 0.5927 | 0.0201 | 0.8003 |

Table 4: Ablation on our memory stores.

| Setting | IoU↑ | LE↓ | CA↑ |
|---|---|---|---|
| All memory stores | **0.5925** | **0.0198** | **0.8006** |
| No ego mem. | 0.5748 | 0.0220 | 0.7837 |
| No exo mem. | 0.5420 | 0.0293 | 0.7495 |

Table 5: Ablation on memory size.

| Memory | IoU↑ | LE↓ | CA↑ |
|---|---|---|---|
| 4 | 0.5921 | 0.0205 | 0.8004 |
| 6 | 0.5925 | 0.0198 | 0.8006 |
| 8 | 0.5963 | 0.0196 | 0.8025 |

mechanism to selectively emphasize and integrate key information from both sources, leading to more effective fusion. In ***appendix***, we also demonstrate the modularity of MV-MoE by integrating it into other backbones.

**Memory stores.** Table 4 summarizes the performance of our model when each memory bank is ablated individually. The results clearly demonstrate the effectiveness of our dual-memory design and underscore the importance of integrating both egocentric and exocentric views to achieve robust perception in complex scenarios.

**Memory compression strategies.** Table 6 compares different memory compression strategies. The FIFO (first-in, first-out) approach stores the most recent N frames in the memory bank. The IoU selection strategy [5] follows a conditional FIFO approach based on IoU scores predicted by SAM 2. In contrast, the cluster selection method first applies average pooling to all frames, followed by clustering to identify and retain the most representative frames. Notably, these methods operate at the frame level, whereas our proposed method performs compression at the feature point level. The results demonstrate that our strategy more effectively aids the model in segmenting objects.

Table 6: Ablation on our memory compression strategy.

| Setting | IoU↑ | LE↓ | CA↑ |
|---|---|---|---|
| FIFO | 0.5823 | 0.0214 | 0.7874 |
| Cluster | 0.5867 | 0.0208 | 0.7942 |
| IoU Sel. | 0.5880 | 0.0204 | 0.7948 |
| Ours | **0.5925** | **0.0198** | **0.8006** |

**Frame number and memory size.** As show in Table 3, increasing the number of frames brings gains although there could be some variance and we use a default of 8 to balance speed and accuracy. Meanwhile, increasing the (maximum) number of memories $M$, generally helps the performance, as in Table 5. We use a default value of 6 past frames to strike a balance between temporal context length and computational cost.

Due to space limitation, we include more results, analysis, and discussion in the ***appendix***. Please kindly refer to our ***appendix*** for details.

## 5 Conclusion

In this paper, we present LM-EEC, a novel framework that builds upon the strong generalization capabilities of SAM 2 to effectively tackle challenges such as severe object occlusions and the presence of numerous small objects. To better exploit multi-view information, LM-EEC introduces a Memory-View Mixture-of-Experts module that dynamically integrates egocentric and exocentric features. In addition, we enhance the memory management mechanism of SAM 2 by constructing a dual long-term memory bank using a view-specific compression strategy, which preserves discriminative information while reducing redundancy. Extensive experiments on the EgoExo4D benchmark show that LM-EEC consistently outperforms prior approaches, especially in complex and cluttered scenes.

**Acknowledgment.** Libo Zhang was supported by National Natural Science Foundation of China (No. 62476266). Heng Fan was not supported by any fund for this work.

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

# Appendix

The appendix is structured as follows:

- **Section A**: We first analyze model performance across diverse activity scenarios to evaluate its generalization capability.

- **Section B**: We then provide qualitative results, comparing the proposed LM-EEC to baselines.

- **Section C**: We discuss the limitation of our model and reflect on its broader impact.

- **Section D**: We also visualize attention maps to demonstrate how our model establishes precise object-level correspondence between egocentric and exocentric views, enabling cross-view alignment.

- **Section E**: We validate long-term ego-exo correspondence using association accuracy over videos of varying lengths.

- **Section F**: We demonstrate the modularity of MV-MoE by integrating it into alternative backbones, showing the architectural transferability.

- **Section G**: We also provide basic training configurations for all compared methods.

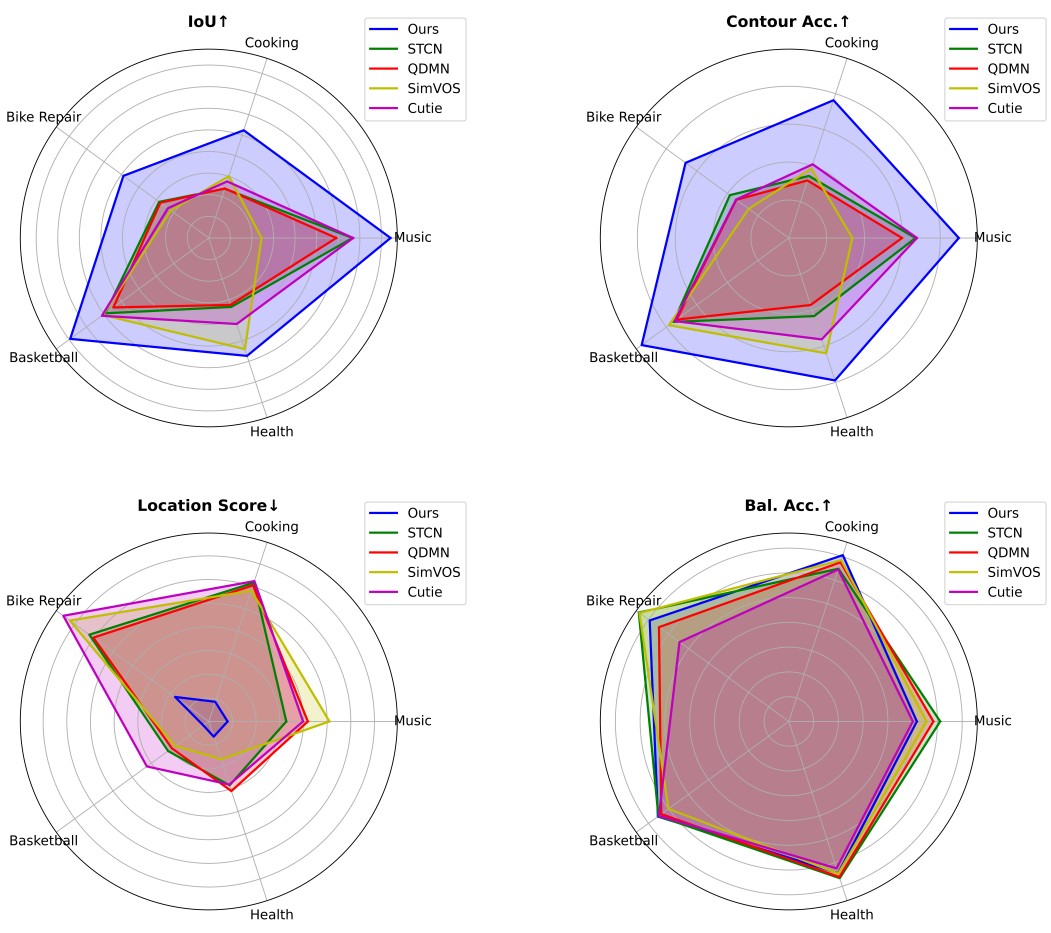

Figure 6: Comparison across different activity scenarios for each model.

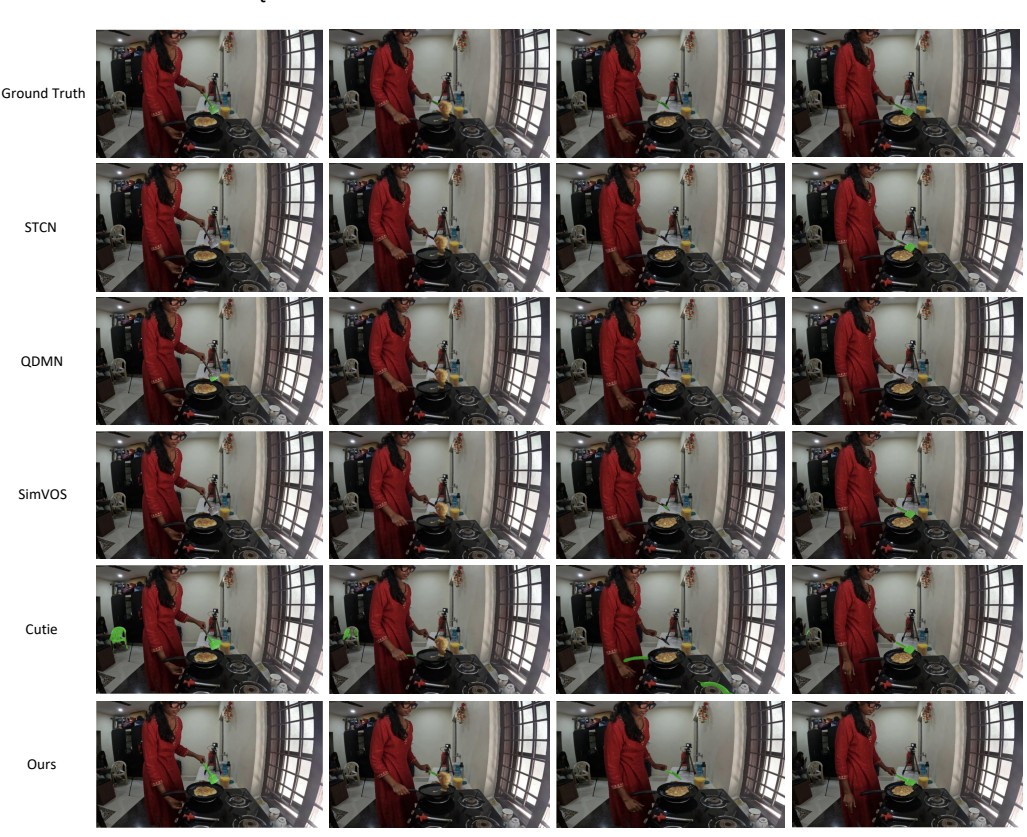

Figure 7: Ego to Exo results of different approaches.

## A  Analysis on Different Scenarios

Beyond object size, we further analyze model performance across different activity scenarios in EgoExo4D, which consists of Cooking, Basketball, Bike Repair, Health, and Music. We compare the results of various models across these scenarios to assess their robustness and adaptability.

As shown in Figure 6, some activities, such as Basketball, are generally easier to model due to limited variation in object shape and appearance. In contrast, activities like Cooking and Bike Repair pose greater challenges, as objects exhibit more diverse appearances and shapes across different views.

Notably, in these more challenging scenarios, our model demonstrates a clear advantage. The performance gap between our method and other baselines is more pronounced in these cases, highlighting the effectiveness of our approach in handling complex and highly variable environments.

## B  Qualitative Results Comparison

We further compare the qualitative results of our model with other baselines. As shown in Figure 7 and 8, while some models struggle to fully segment the object (e.g., the stainless steel spatula) or to distinguish it from similar objects in the background, our model accurately segments the object and effectively handles occlusion and variations in its appearance.

## C  Limitation and Broader Impact

**Discussion of Limitation.** As observed, our model does not achieve the highest performance in terms of balanced accuracy. This indicates that when the target object disappears from the scene, the

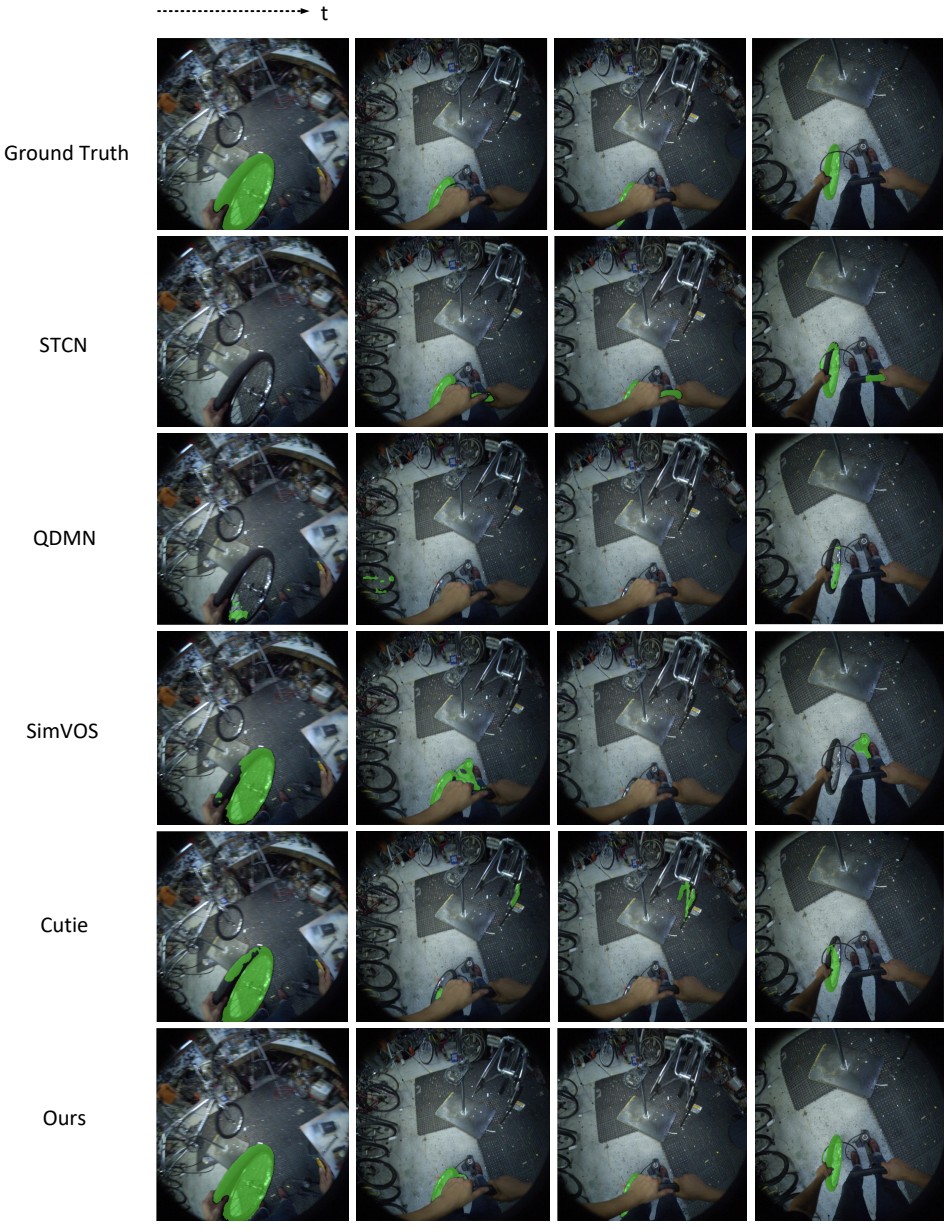

Figure 8: Exo to Ego results of different approaches.

model may mistakenly segment visually similar background objects. Representative cases of such failures are illustrated in Figure 9.

This limitation primarily stems from the fact that existing baselines do not explicitly consider object disappearance during training. As a result, the model tends to associate and segment visually similar objects in the current frame, even when the target object is no longer present. This observation reveals an inherent trade-off among IoU, contour accuracy, location score, and balanced accuracy.

In this work, we build upon SAM 2 as the baseline architecture to develop our model. While SAM 2 demonstrates strong matching capability and performs effectively in video segmentation, it lacks explicit mechanisms to handle object disappearance. Consequently, our model inevitably inherits this limitation to some extent. We believe that incorporating targeted data augmentation during training or

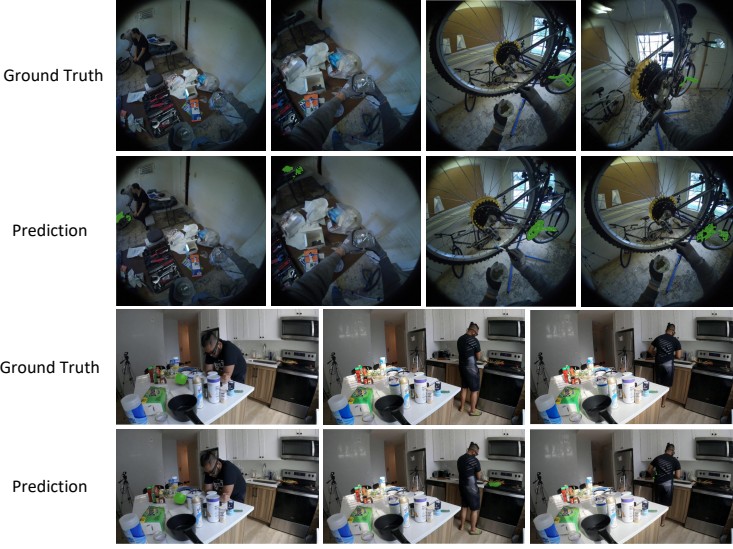

Figure 9: Failure cases.

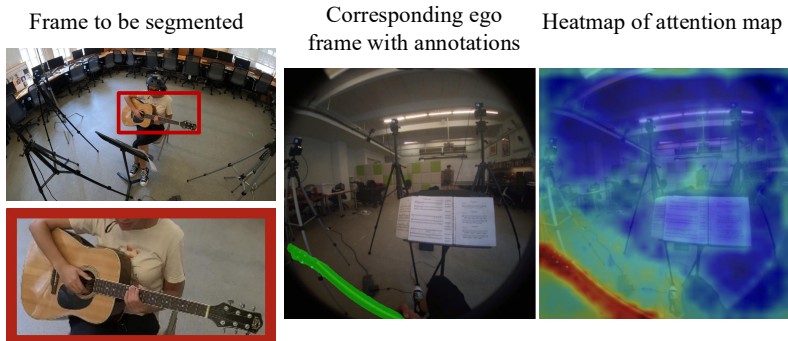

Figure 10: Attention map visualization.

introducing an additional branch to explicitly predict object presence or absence could be promising directions to address this issue in future work.

**Discussion of Broader Impact.** The proposed LM-EEC which focuses on establishing object-level correspondence between egocentric and exocentric views has the potential to benefit a wide range of real-world applications, including intelligent robotics, augmented reality (AR), and assistive technologies. By enhancing the ability of AI systems to reason about objects from multiple viewpoints, LM-EEC could contribute to more accurate and intuitive human-AI interaction, such as guiding users through complex tasks or supporting remote collaboration.

It's worth noting that our work carries potential risks, particularly concerning privacy and surveillance, as the ability to associate first-person and third-person views could be misused for unauthorized tracking or monitoring of individuals beyond intended applications. Therefore, we emphasize that our work must be strictly applied within ethical and privacy-compliant contexts.

## D   Attention Map Visualization

In addition, we visualize the attention map of both the current frame and the corresponding annotated frame. Specifically, we display the attention map related to the current target to be segmented from

Table 7: Association accuracy of our method and the baseline under different video lengths.

| Video Length | <200 frames | 200–600 frames | >600 frames |
|---|---|---|---|
| Baseline Model | 0.5580 | 0.6260 | 0.4724 |
| Our Model | **0.6054** | **0.6732** | **0.6150** |

an alternate view. As shown in Figure 10, our model accurately matches objects from both ego and exo views (e.g., the guitar).

# E   Long-term Correspondence Evidence

To verify the effectiveness of our method in modeling long-term ego-exo correspondence, we conduct an experiment to evaluate the **association accuracy** across videos of different lengths. Specifically, we compare our model with the baseline (i.e., SAM 2) that excludes the core MV-MoE and memory compression modules.

In this experiment, the **association accuracy** is defined as the ratio of frames where the Intersection over Union (IoU) between the predicted and ground truth masks exceeds a threshold of 0.5. The comparison results are summarized in Table 7.

As shown in Table 7, our model consistently outperforms the baseline across all video lengths. Notably, for longer videos with more than 600 frames, our method achieves a significant improvement of approximately **14%** in association accuracy compared with the baseline model. This result effectively validates the capability of our approach to maintain robust **long-term ego-exo correspondence** over extended temporal durations.

# F   Modularity of MV-MoE

While the proposed MV-MoE is initially designed on top of the SAM 2 architecture, its core component, i.e., the dual-branch routing mechanism that adaptively assigns contribution weights to each expert feature along both channel and spatial dimensions, is conceptually modular and can be readily integrated into other architectures that involve multi-source feature fusion. To demonstrate the generality of MV-MoE, we conduct additional experiments by integrating it into two alternative frameworks with different segmentation bones, including STCN and QDMN, for the Ego-Exo Correspondence task. The results on the Ego2Exo test split of the EgoExo4D dataset are shown in Table 8.

From Tab. B, we can clearly observe that, incorporating MV-MoE consistently enhances performance across metrics, indicating that the module is not only effective but also transferable to different backbone architectures. This suggests that MV-MoE can be easily adapted to other segmentation architectures/backbones and future versions of SAM 2-like frameworks.

Table 8: Application of MV-MoE on other segmentation backbones.

| Model | IoU $\uparrow$ | LE $\downarrow$ | CA $\uparrow$ | BA $\uparrow$ |
|---|---|---|---|---|
| STCN | 27.39 | 0.109 | 0.378 | 61.97 |
| STCN + MV-MoE | **30.51** | **0.083** | **0.436** | **62.81** |
| QDMN | 27.03 | 0.108 | 0.346 | 63.16 |
| QDMN + MV-MoE | **28.75** | **0.100** | **0.376** | **64.02** |

# G   Basic Training Configurations

All baselines were implemented from official code with minimal changes for EgoExo4D, following original training settings. Table 9 summarizes the training configurations of our method and other approaches. Notably, all the compared method are trained for more epochs/iterations than our

proposed model to ensure sufficient optimization. Our goal is to ensure fair and optimal performance for all methods on EgoExo4D.

Table 9: Training configurations of compared methods.

| Model | Epochs (Iterations) | Optimizer |
|---|---|---|
| XSegTx | 250 (250K iter.) | Adam |
| XView-Xmem (w/ finetuning) | 358 (100K iter.) | AdamW |
| XView-Xmem (+ XSegTx) | 358 (100K iter.) | AdamW |
| STCN | 258 (50K iter.) | Adam |
| QDMN | 207 (80K iter.) | Adam |
| GSFM | 213 (55K iter.) | Adam |
| SimVOS | 193 (300K iter.) | Adam |
| Cutie | 167 (125K iter.) | AdamW |
| LM-EEC(Ours) | 60 (12K iter.) | AdamW |

