# OpenReview forum: "Robust Ego-Exo Correspondence with Long-Term Memory"
_NeurIPS.cc/2025/Conference — NeurIPS 2025 poster_

### Official Review · Reviewer_2TFe · 2025-07-01

**Clarity:** 3
**Significance:** 3
**Originality:** 3
**Rating:** 4
**Confidence:** 3

**Summary:**

This paper introduces LM-EEC, a novel framework for establishing object-level correspondence between egocentric (first-person) and exocentric (third-person) videos by enhancing SAM 2 with two key innovations: (1) a Memory-View Mixture-of-Experts module that adaptively fuses cross-view features using channel and spatial attention, and (2) a dual-memory bank system with compression to preserve long-term object context. Evaluated on the EgoExo4D benchmark, LM-EEC achieves state-of-the-art results, outperforming methods like XSegTx and XView-XMem, particularly in handling small objects and viewpoint variations.

**Questions:**

See in Weaknesses.

**Ethical Concerns:**

["Major Concern: Data privacy, copyright, and consent"]

**Final Justification:**

Thanks for the response. I have no further questions. I will keep the score.

**Limitations:**

Yes.

**Quality:**

3

**Strengths And Weaknesses:**

Paper strengthens:

(1) This paper introduces LM-EEC, a novel framework enhancing SAM 2 for ego-exo correspondence. It proposes Memory-View Mixture-of-Experts (MV-MoE) for adaptive feature fusion across views, addressing feature distribution gaps, and designs a dual-memory bank system with view-specific compression to retain long-term information efficiently.

(2) In comprehensive ablation studies, validate each component (MV-MoE, dual memory, compression). This work outperforms 10+ baselines (e.g., XSegTx, XView-XMem, STCN) by significant margins (e.g., +18% IoU in Exo2Ego tasks).

(3) It solves critical challenges: extreme viewpoint variations, occlusions, small objects, and long-video redundancy. Meanwhile, it achieves SOTA results on the challenging EgoExo4D benchmark, demonstrating real-world applicability (e.g., AR, robotics).

(4) Well-structured with clear task definitions (Ego2Exo/Exo2Ego), SAM 2 adaptations, and visualizations (Figures 1-5).

Paper weakness:

(1) In Table 3 (Section 4.4), the paper reports inconsistent performance when varying training frame numbers, showing degraded IoU (0.5873) at 10 frames compared to 8 frames (0.5882). The authors fail to provide ablation studies on regularization strategies or frame sampling methods that could explain this anomaly, undermining their claims about optimal memory configuration.

(2) The description of the memory encoder in Section 3.3 contains critical gaps in technical specification. While the paper repeatedly references an "off-the-shelf memory encoder" for processing features before storage in the dual-memory system, it completely omits essential architectural details, including layer types, parameter counts, and input/output dimensions.

(3) The compression approach described in Section 3.5 (Equations 8-10) demonstrates concerning conceptual simplicity. The method relies entirely on Euclidean distance thresholds between adjacent frames followed by naive feature averaging, without any consideration of more sophisticated techniques like attention-based pooling or entropy analysis to preserve critical information. This is particularly problematic for handling non-adjacent critical events like object reappearance after occlusion, as shown in the paper's own Figure 4. The authors provide no quantitative evidence that their compression method actually preserves the most discriminative features, despite making strong claims about long-term information retention.

(4) The comparison with baseline methods in Table 1 (Section 4.3) suffers from serious methodological shortcomings. While the paper states that all models were "trained or fine-tuned on the EgoExo4D dataset," it provides no details about the fine-tuning protocols used-including number of epochs, optimizer settings, or data splits. This lack of transparency makes it impossible to verify whether the baseline implementations achieved their full potential performance, maybe it’s better show more details of the implementation of various comparison methods.

(5) Despite bold assertions in the Abstract about "strong generalization across diverse scenarios," the paper provides no evidence beyond single-dataset evaluation on EgoExo4D. The complete absence of cross-dataset validation (e.g., on EPIC-KITCHENS or other standard benchmarks) renders these generalization claims entirely speculative. This is particularly concerning for a method that purports to handle challenging real-world conditions like viewpoint variation and occlusion conditions that can vary significantly across different datasets and environments.

(6) The paper's treatment of occlusion challenges reveals significant limitations. While Section 1 emphasizes handling "severe object occlusions" as a key contribution, the actual performance gains in Balanced Accuracy (BA) metrics are minimal-just +2.66% for Ego2Exo and +1.03% for Exo2Ego compared to the base model (Table 1). Moreover, the qualitative results in Figure 1 conspicuously avoid showing challenging occlusion cases, focusing instead on relatively straightforward examples where objects remain largely visible. This selective presentation undermines the paper's claims about robust occlusion handling.

---

> ### Author Rebuttal · Authors · 2025-07-31
>
> We sincerely thank the reviewer for the thoughtful and detailed review. Below, we address all concerns point-by-point.
>
> > **Q1**: Table 3 (Section 4.4) shows inconsistent performance—IoU drops at 10 frames (0.5873) compared to 8 frames (0.5882)—without ablation on regularization or frame sampling.
>
> **A1**:
> Thanks for this careful comment. Upon further analysis, we believe the performance degradation when increasing the number of training frames from 8 to 10 is likely due to the introduction of redundant or noisy information. Specifically, the additional frames may not always provide complementary cues, but instead introduce irrelevant context or conflicting temporal signals that interfere with effective learning. This phenomenon is not uncommon in video object segmentation tasks, as similarly observed in prior works such as “End-to-End Video Instance Segmentation with Transformers” and “Segment Anything in Images and Videos (SAM 2)”.
>
> Tab. C: Results of the random sampling strategy.
> | Frame | IoU ↑  | LE ↓   | CA ↑   |
> |-------|--------|--------|--------|
> | 8     | 0.5894 | 0.0199 | 0.7967 |
> | 10    | 0.5905 | 0.0202 | 0.7997 |
>
> As suggested, to further explain this anomaly, we conduct an ablation study on the frame sampling strategy. Different from the adjacent frame sampling method in the paper, we utilize a random frame sampling strategy to increase temporal diversity. The experimental results are shown in Tab. C. From Tab. C, we can see, similar to our used adjacent frame sampling method (see Tab. 3), when increasing frames from 8 to 10, the performance slightly decreases, which is consistent with our above analysis.
>
> Thanks again, we will include the above analysis and results in the revision.
>
> > **Q2**: Section 3.3 lacks key technical details about the “off-the-shelf memory encoder,” such as its architecture, layer types, and input/output dimensions.
>
> **A2**:
> Thanks for this helpful comment. Here we provide the architectural details mentioned by the reviewer.  ***(i) Layer Types***: The memory encoder consists of two main components: (1) MaskDownSampler, which processes the binary object mask before feature fusion; and (2) Fuser, a ConvNeXt-style fusion block that integrates pixel-level visual features with the downsampled mask signal. ***(ii) Parameter Count***: The total number of trainable parameters in the memory encoder is 5.3 M. ***(iii). Input/Output Dimensions***: The memory encoder takes a visual feature map of shape [B, 256, 30, 30] and a binary mask of shape [B, 1, 480, 480] as input, and outputs a fused embedding of shape [B, 64, 30, 30], where B=4 during training and B=1 during testing.
>
> We will incorporate the above details in the revision. Our code will also be released. Again, thanks!
>
> > **Q3**: **(a)** The compression method in Section 3.5 is overly simplistic, using only Euclidean distance and feature averaging, with no advanced techniques to retain more critical information—especially for non-adjacent events like reappearance after occlusion. **(b)** The authors provide no quantitative evidence that their compression method actually preserves the most discriminative features, despite making strong claims about long-term information retention.
>
> **A3**:
> Thanks for the insightful comment.
>
> **To sub-question (a):** Our current goal is to develop a simple, efficient, and effective memory compression strategy to maintain long-term information for ego-exo correspondence. For this purpose, we design the proposed memory compression method using Euclidean similarity. Despite its simplicity, it enables efficient training and inference while preserving useful information for segmentation. Compared to the FIFO strategy, our method consistently improves IoU by ~1%, contour accuracy by ~1.6%, and reduces location error from 0.0276 to 0.0258 (see Tab. 6), demonstrating its effectiveness.
>
> However, we agree that more advanced techniques could better preserve critical features, especially for non-adjacent key events like reappearance after occlusion. Such improvements may further enhance segmentation in complex scenarios, and we leave this exploration to future work.
>
> **To sub-question (b):** To further show quantitative evidence that our comprehension method helps retain discriminative long-term information, we conduct a quantitative comparison between the baseline model (i.e., SAM 2) and a variant using our compression module. Specifically, we evaluate ***association accuracy***—the proportion of frames with IoU > 0.5—under different video length as a proxy for the model’s ability to maintain consistent object tracking and segmentation over time. Results are shown in Tab. D.
>
> Tab. D: Quantitative analysis of our memory compression method via association accuracy on Ego2Exo.
> | Video Length            | <200 frames | 200-600 frames | >600 frames |
> |-------------------------|-------------|----------------|-------------|
> | Baseline                | 0.5580      | 0.6260         | 0.4724      |
> | Baseline w/ compression | 0.5646      | 0.6381         | 0.5385      |
>
> Tab. D shows our compression method consistently improves association accuracy, especially on longer videos (>600 frames) with over 6% gain, demonstrating effective retention of long-term information.
>
> Thanks again! We will include the above analysis in revision.
>
> > **Q4**: The baseline comparison in Table 1 lacks transparency—key fine-tuning details (e.g., epochs, optimizer, data splits) are omitted, making it unclear whether baselines reached their full potential.
>
> **A4**: Thanks for this comment. Here we clarify the implementation details of other compared methods.
>
> ***(i) Training Configurations***: All baselines were implemented from official code with minimal changes for EgoExo4D, following original training settings. Tab. E summarizes the training configurations of our method and other approaches. Notably, all the compared method are trained for ***more*** epochs/iterations than our proposed model to ensure sufficient optimization. Our goal is to ensure fair and optimal performance for all methods on EgoExo4D.
>
> Tab. E: Summary of training configurations of different methods.
> | Model                      | Epochs (Iterations)     | Optimizer |
> |----------------------------|-------------------------|-----------|
> | XSegTx                     | 250 (250K iter.)        | Adam      |
> | XView-Xmem (w/ finetuning) | 358 (100K iter.)        | AdamW     |
> | XView-Xmem (+ XSegTx)      | 358 (100K iter.)        | AdamW     |
> | STCN                       | 258 (50K iter.)         | Adam      |
> | QDMN                       | 207 (80K iter.)         | Adam      |
> | GSFM                       | 213 (55K iter.)         | Adam      |
> | SimVOS                     | 193 (300K iter.)        | Adam      |
> | Cutie                      | 167 (125K iter.)        | AdamW     |
> | Ours                       | 60 (12K iter.)          | AdamW     |
>
> ***(ii) Data Splits***: We follow the official data split of EgoExo4D, which contains 756 takes for training, 202 takes for validation, and 291 takes for testing. These splits were applied consistently across all methods for fair comparison.
>
> We will include the above clarifications in revision. Besides, we will release the implementations of all methods. Again, thanks!
>
> > **Q5**: The paper claims strong generalization but only reports results on EgoExo4D, without cross-dataset validation (e.g., EPIC-KITCHENS), making the generalization unsubstantiated.
>
> **A5**: Thank you for the insightful comment. We use EgoExo4D because it is the only dataset that provides ego-exo correspondence annotations (i.e., object masks in both views), and it is also large and diverse. Notably, the ego-exo correspondence task was first proposed in EgoExo4D.
>
> We agree that broader evaluation is important. And we have carefully reviewed the EPIC-KITCHENS dataset. We found that, the EPIC-KITCHENS dataset mainly focuses on the egocentric perspective and do not offer dense segmentation mask annotations. Therefore, it is not directly applicable to evaluate our method. We also surveyed other datasets but found none supporting ego-exo correspondence.
>
> That said, we appreciate the suggestion and see evaluation on more datasets as a valuable direction. In our future work, we plan to develop a ***new*** dataset, as a ***supplement*** to EgoExo4D, by featuring more complex outdoor environments and broader activity domains, to better assess generalization.
>
> Thanks again! We will include this clarification in revision.
>
> > **Q6**: **(a)** The paper emphasizes occlusion handling but shows only marginal BA gains (+2.66%, +1.03%) **(b)** and avoids hard occlusion cases in Figure 1, weakening its claims of robustness.
>
> **A6**: Thanks for this insightful comment.
>
> **To sub-question (a):** Balanced Accuracy (BA) evaluates the model’s ability to detect the presence or absence of the target in a frame. In contrast, occlusion refers to cases where the target is present but partially or heavily blocked. Thus, occlusion handling is better reflected in metrics like IoU, location error, and contour accuracy, where our method shows clear improvements over baselines.
>
> **To sub-question (b):** Sorry for the confusion. Fig. 1 illustrates overall performance comparisons, not specifically occlusion. Occlusion cases are shown in Fig. 7 and Fig. 10 (e.g., a shovel occluded by eggs, a tire blocked by a hand). To further substantiate our claims, we will add more visualizations focusing on occlusion in the revision. Again, thanks!
>
> > **Q7**: Ethical Concerns: Data privacy, copyright, and consent
>
> **A7**:
> Thank you for raising this important concern. We use the EgoExo4D dataset under an official research license from the University of Bristol, with proper attribution. We confirm that all usage complies with the license and no additional personally identifiable information is involved.
>
> Once again, thanks for the helpful feedback. We will revise the paper to enhance clarity.

---

> > ### Comment · Reviewer_2TFe · 2025-08-07
> > **after rebuttal**
> >
> > Thanks for the response. I have no further questions.

---

> > > ### Author Response · Authors · 2025-08-07
> > > **Thanks for your reply**
> > >
> > > Dear Reviewer
> > >
> > > Thank you again for your comments and suggestions to improve our manuscript. We will include all points in revision. Again, thank you very much.

---

### Official Review · Reviewer_dkHk · 2025-07-03

**Clarity:** 3
**Significance:** 3
**Originality:** 2
**Rating:** 4
**Confidence:** 4

**Summary:**

This paper proposes a new framework for the ego-exo object correspondence task. Given a pair of synchronized ego-exo videos and a sequence of query masks for an object of interest in one of the videos, the objective of the task is to identify the corresponding masks of the same object in each synchronized frame of the other view. The proposed framework adapts the Segment Anything Model 2 (SAM2) to this cross-view segmentation task, and develops two new modules, a memory-view mixture-of-experts module and a dual-memory bank system to enhance cross-view feature fusion and long-term memory management. Experiments on the EgoExo4D benchmark show that the proposed method achieves state-of-the-art results.

**Questions:**

1. The use of term “mixture-of-experts” may cause confusion. Often the term of expert refers to some independent models, but in this paper it is used for the features from certain dimension (channel or spatial).
2. Brief introduction (e.g., data amount, data split) of the object correspondence benchmark from Ego-Exo4d dataset would help the reader comprehends the task but is not given either.

**Ethical Concerns:**

["NO or VERY MINOR ethics concerns only"]

**Final Justification:**

Most of my concerns have been addressed by the rebuttal, and I have no major concern. Thus I recommend acceptance.

**Limitations:**

yes

**Paper Formatting Concerns:**

NA.

**Quality:**

2

**Strengths And Weaknesses:**

**Strengths**
1) *Clarity*: The paper is well-written and easy to follow. The motivation and idea are clearly stated and the details of method design is well explained.
2) *Significance*: The paper adapts the well-known SAM2 to the ego-exo object correspondence task with two newly developed modules, and significantly outperforms existing baselines of video object segmentation, largely advancing the state-of-the-art performance.
3) *Originality*: The proposed Memory-View Mixture-of-Experts (MV-MoE) module and the dual-memory compression strategy are reasonable and validated by the experiments.

**Weaknesses**
1. The proposed method is largely based on the segmentation framework of SAM2, and the main technical novelties come from the two proposed module of MV-MoE and dual-memory compression. However, MV-MoE is essentially the combination of channel attention and spatial attention which are common (or even “old-fashioned”) design of attention modeling. Dual-memory compression is also simply designed and the improvement over SAM2’s memory design is marginal (less than 1% of IoU in Table 6). Thus the overall technical novelty seems to be of limited significance.
2. The paper introduces SAM2 to the object correspondence task and largely advance the SOTA performance, which is worth of praise. However, as far as my understanding, the performance improvement largely owes to the strong base model of SAM2. The base model already outperforms related methods for most metrics (IoU, LS, CA).
3. Some important implementation details are missing. It is not clarified which parts of the framework are updated during training and which are frozen. The data amount, training time cost, and inference efficiency are not given either. Since the memory compression strategy relies on the computation of feature similarity over all temporal and spatial location in the memory bank, it may lowers the inference speed.

---

> ### Author Rebuttal · Authors · 2025-07-31
>
> Thanks for the thoughtful and detailed feedback. We sincerely appreciate the reviewer's recognition of the clarity, significance, and originality of our work in adapting SAM 2 to the ego-exo object correspondence task through the introduction of two novel modules. Below, we provide point-by-point responses to the concerns raised.
>
> > **Q1**: The proposed method is largely based on the segmentation framework of SAM2, and the main technical novelties come from the two proposed module of MV-MoE and dual-memory compression. However, MV-MoE is essentially the combination of channel attention and spatial attention which are common (or even "old-fashioned") design of attention modeling. Dual-memory compression is also simply designed and the improvement over SAM2's memory design is marginal (less than 1% of IoU in Table 6). Thus the overall technical novelty seems to be of limited significance.
>
> **A1**: We appreciate the reviewer's perspective and would like to further clarify the novelty and significance of our contributions:
>
> **(i) About the MV-MoE module**: We understand the reviewer's concern regarding the "old-fashioned" nature of channel and spatial attention. While the attention is common, they are ***under-explored*** in the ego-exo correspondence task for representation learning. Our ***contribution*** lies not in simply reusing these mechanisms, but in how they are jointly leveraged within a modular mixture-of-experts framework specifically designed for multi-feature fusion to improve ego-exo correspondence. The experiments reported in our paper effectively validate that the proposed MV-MoE is crucial for performance improvement.
>
> **(2) About the dual-memory compression**: Thanks for the comment. Our goal is to develop a simple, efficient, and effective memory compression strategy to main long-term information for ego-exo correspondence. For this purpose, we design the proposed memory compression method using Euclidean similarity. Despite its simplicity, it allows our model to effectively maintain useful long-term information for segmentation and execute efficiently in training and inference. Using our dual-memory compression, the improvement in IoU is around 1%, which is non-trivial when using the rigorous IoU metric on the whole video. In addition, we show that it clearly improves the contour accuracy (CA) with around 1.6% and can decrease the location error (LE) from 0.0276 to 0.0258 compared to the FIFO strategy, showing its efficacy in consistently improving ego-exo correspondence performance.
>
>
> We will incorporate the above clarifications in the revision to make our contributions more clear. Again, thanks!
>
>
> > **Q2**: The paper introduces SAM2 to the object correspondence task and largely advance the SOTA performance, which is worth of praise. However, as far as my understanding, the performance improvement largely owes to the strong base model of SAM2. The base model already outperforms related methods for most metrics (IoU, LS, CA).
>
> **A2**: Thanks for this careful comment. SAM 2 is chosen as our baseline because of its strong generalization ability. Yes, as noted by the reviewer, SAM 2 itself already shows impressive results by outperforming related methods, but directly applying SAM 2 to the exo-ego correspondence faces several challenges, such as suboptimal feature fusion across ego and exo views and ineffective long-term memory maintenance. Our proposed method addresses these challenges by introducing two key contributions, including the MV-MoE module for improved feature integration, and a dual-memory compression mechanism to manage memory more effectively. On the already-powerful SAM 2 baseline, our method still improves its performance by a large margin with 8.5% and 2.9% IoU gains in Exo2Ego and Ego2Exo settings (please see Tab. 1 in our paper). These substantial performance gains, achieved over an already powerful baseline, clearly demonstrate the efficacy and value of our proposed method in improving ego-exo correspondence.
>
> We will include the above clarification in the revision. Again, thanks!
>
> > **Q3**: Some important implementation details are missing. It is not clarified which parts of the framework are updated during training and which are frozen. The data amount, training time cost, and inference efficiency are not given either. Since the memory compression strategy relies on the computation of feature similarity over all temporal and spatial location in the memory bank, it may lower the inference speed.
>
> **A3**: Sorry for the confusion. Here we provide clarifications for the implementation details mentioned by the reviewer.
>
> ***(i) Trainable Components***: Given the large-scale nature of EgoExo4D, we train all modules jointly based on the pre-trained SAM2 checkpoint, without freezing any components.
>
> ***(ii) Data Volume***: The EgoExo4D dataset contains 5,500 annotated objects across 1,249 ego-exo video pairs, with around 4 million annotated frames. This results in 742K egocentric and 1.1M exocentric segmentation masks. We adopt the official dataset split: 756 takes for training, 202 for validation, and 291 for testing.
>
> ***(iii) Training Cost***: The training process takes approximately 13 hours on 8xA100 GPUs with a batch size of 4 for 60 epochs.
>
> ***(iv) Inference Efficiency***: Inference is conducted on a single V100 GPU at ~8.4 FPS, compared to ~9.0 FPS for the baseline SAM 2. The proposed memory compression strategy slightly reduces inference speed but yields improved performance, which we believe is a worthwhile trade-off.
>
> We will include the above details explicitly in the revision to improve clarity of our method. Our full code and model will be released for reproducibility. Again, thanks for this helpful comment!
>
> > **Q4**: The use of term "mixture-of-experts" may cause confusion. Often the term of expert refers to some independent models, but in this paper it is used for the features from certain dimension (channel or spatial).
>
> **A4**:
> Thanks for this careful comment. We agree and understand the reviewer's concern. As suggested, to avoid potential confusion, we will modify the term "mixture-of-experts" in our method in the revision. Again, thanks!
>
>
> > **Q5**: Brief introduction (e.g., data amount, data split) of the object correspondence benchmark from Ego-Exo4d dataset would help the reader comprehends the task but is not given either.
>
> **A5**: Thanks for this comment. The EgoExo4D dataset contains 5,500 annotated objects across 1,249 ego-exo video pairs, with around 4 million annotated frames. This results in 742K egocentric and 1.1M exocentric segmentation masks. We adopt the official dataset split in our experiment, where 756 videos are used for training, 202 for validation, and 291 for testing. We will include this introduction of the EgoExo4D dataset in the revision. Thanks again!
>
>
> Once again, we deeply appreciate the reviewer's constructive feedback. We have taken all suggestions seriously and would revise the paper to improve clarity, rigor, and transparency.

---

> > ### Comment · Reviewer_dkHk · 2025-08-08
> >
> > Thanks for the detailed responses by the authors. Most of my concerns have been addressed, and I would maintain my positive rating. Please revise the paper accordingly.

---

> > > ### Author Response · Authors · 2025-08-08
> > > **Thanks for your reply**
> > >
> > > Dear Reviewer,
> > >
> > > Thank you very much for your constructive and helpful comments and feedback on our work. We will include all points mentioned in our rebuttal in the revision. Again, thank you very much.

---

### Official Review · Reviewer_r7ib · 2025-07-03

**Clarity:** 3
**Significance:** 3
**Originality:** 3
**Rating:** 5
**Confidence:** 5

**Summary:**

This paper addresses the challenging problem of object-level correspondence across egocentric and exocentric views, which an important problem with applications in AR guidance, robotic perception, etc. The core difficulty lies in large viewpoint changes, object occlusions, and scale variance.

To tackle this, the authors propose **LM-EEC**, a SAM2-based video object segmentation framework that introduces:
- **Memory-View Mixture-of-Experts (MV-MoE)**: A dual-branch dynamic fusion module that adaptively weighs memory-aware and view-specific features across channel and spatial dimensions using learned routing mechanisms.
- **Dual Memory Banks with Compression**: Separate memory banks for ego and exo views, with a lightweight temporal feature compression strategy to reduce redundancy and preserve long-term dependencies efficiently.

The model is benchmarked on the EgoExo4D dataset and achieves strong improvements over both prior SOTA methods (e.g. XSegTx, XView-XMem, SimVOS) and the SAM2 baseline across multiple metrics (IoU, location error, contour accuracy, etc.).

**Questions:**

Please see the weaknesses listed above, which also contains a few questions for the authors.

**Ethical Concerns:**

["NO or VERY MINOR ethics concerns only"]

**Final Justification:**

This is a strong paper, with interesting algorithmic components and satisfactory experiments. The authors' rebuttal has addressed my concerns, especially about the dataset and application of the MV-MoE module on other segmentation backbones. The newly added experiments confirm the author's hypothesis that the MV-MoE is indeed useful for the Ego-Exo task.

I maintain my rating at "Accept".

**Limitations:**

Yes

**Quality:**

3

**Strengths And Weaknesses:**

# Strengths
1. **Well-motivated technical design**: The use of dual memory banks and expert-weighted fusion directly addresses the issues of feature misalignment and memory inefficiency that plague SAM2 when applied to the Ego-Exo task.
2. **Novel implementation**: The MV-MoE module integrates both spatial and channel attention in a residual fashion, providing fine-grained expert weighting and avoiding simple addition-based prompt fusion.
3. **Efficient memory compression**: The memory pruning strategy is based on pointwise Euclidean redundancy, which is simple yet effective, and goes beyond frame-level heuristics like FIFO or clustering.
4. **Experiments**:
    - The paper clearly isolates the contribution of each component - MV-MoE, memory compression, and training frame length - showing consistent improvement in isolation and combination.
    - The method outperforms previous methods in IoU, contour accuracy, and location error across both Ego2Exo and Exo2Ego settings, and across small/large object subsets.
    - Qualitative analysis and attention visualizations further support the method’s robustness to appearance variations and viewpoint shifts.

# Weaknesses
1. **Limited generalization claims. More datasets?**: Despite strong performance on EgoExo4D, the model is evaluated on a single dataset. Claims about generalization could be strengthened with tests on other multi-view video datasets. For example, how about applying this method to **Assembly101** dataset (CVPR 2022) which has multi-view videos for various activities?
    - There's also the dataset from "Replay: Multi-modal Multi-view Acted Videos for Casual Holography", which can be used for evaluation.

2. **Worse performance in Balanced accuracy**: The method underperforms in balanced accuracy compared to XSegTx, indicating poor modeling of object disappearance (i.e., false positives when objects are out of view). *This is also noted by the authors*.

3. **Compression heuristic could be stronger?**: The proposed memory compression merges features purely based on Euclidean similarity. While efficient, it lacks semantics (e.g., object-aware grouping or scene dynamics).

4. **Modularity of MV-MoE**: The fusion module is tightly coupled with SAM2’s architecture. It's not entirely clear how portable or reusable the module is with other segmentation backbones. This is not a big weakness, although it would be nicer is the module was such that it could be used with future versions of SAM2-like architectures.

---

> ### Author Rebuttal · Authors · 2025-07-31
>
> We sincerely thank the reviewer for all the detailed and insightful comments. We truly appreciate the recognition of our technical contributions-including the MV-MoE design, the dual-memory structure, and the efficient compression strategy-as well as the positive assessment of the rigor of our experimental evaluations. Below, we provide point-by-point responses to all the concerns.
>
> > **Q1**: **Question about limited generalization claims. More datasets?**: Despite strong performance on EgoExo4D, the model is evaluated on a single dataset. Claims about generalization could be strengthened with tests on other multi-view video datasets. For example, how about applying this method to **Assembly101** dataset (CVPR 2022) which has multi-view videos for various activities?
> > - There's also the dataset from "Replay: Multi-modal Multi-view Acted Videos for Casual Holography", which can be used for evaluation.
>
> **A1**: Thanks for this comment. The reason why we only conduct experiments on EgoExo4D is because it is the only one that provides unique fine-grained ego-exo correspondence annotations (i.e., object masks in both ego and exo views). In addition, EgoExo4D itself is a large and diverse dataset that contains numerous scenarios. Please note that, the ego-exo correspondence task is firstly proposed in the EgoExo4D dataset.
>
> We understand the reviewer's concern and fully agree with the reviewer that broader evaluation is essential to support claims of generalization. We have carefully reviewed the datasets the reviewer suggested. We found that, although the Assembly101 dataset provides multi-view videos, it primarily targets **action recognition and understanding** tasks (e.g., action recognition, anticipating actions, temporal action segmentation, hand-object interaction, etc.), with annotations limited to action labels and 3D hand poses, but it **lacks the dense object-level mask annotations and correspondence** which are necessary for our ego-exo correspondence task. Similarly, Replay focuses on multi-modal, multi-view synthesis of dynamic content, rather than dense segmentation in synchronized ego-exo views. As such, these datasets are ***not*** directly suitable for evaluating ego-exo correspondence in our method.
>
> That said, we appreciate the reviewer's suggestion and agree that extending evaluation to broader contexts is a valuable direction. As part of our future work, we plan to develop a ***new*** dataset, as a ***supplement*** to the existing EgoExo4D, by featuring more complex outdoor environments and broader activity domains beyond those captured in EgoExo4D, in order to further assess the generalization capabilities of our approach and future research.
>
> Thanks again for this helpful comment. We will include the above clarification in the revision.
>
> > **Q2**: **Question about performance in Balanced accuracy**: The method underperforms in balanced accuracy compared to XSegTx, indicating poor modeling of object disappearance (i.e., false positives when objects are out of view). This is also noted by the authors.
>
> **A2**:
> Thanks for this careful comment. As discussed in the manuscript (Line 435-439 in paper), there exists an inherent trade-off among IoU, contour accuracy, location score, and balanced accuracy. Specifically, our model tends to segment the corresponding objects in the current frame, even when the target objects are not present. This tendency can lead to a higher false positive rate in scenarios involving object disappearance, negatively affecting balanced accuracy. This issue is particularly pronounced in the EgoExo4D dataset due to extreme viewpoint variations and frequent occlusions, resulting in numerous frames where the target object is absent. Interestingly, we observe that models with relatively weaker segmentation capability sometimes achieve higher balanced accuracy in such challenging scenarios, as they tend to refrain from predicting any segmentation in uncertain cases—an effect illustrated in Fig. 7 in the appendix, where SimVOS, QDMN, and STCN show such behavior.
>
> To improve our balanced accuracy, a possible and simple solution is to include a branch in our model to directly predict object presence or absence. However, since our current primary goal is to enhance object-level correspondence and effectively retain informative long-term content. We leave developing specific mechanisms for improving balance accuracy to our future work.
>
> Again, thanks for this helpful comment. We will include the above clarification in the revision.
>
> > **Q3**: **Compression heuristic could be stronger?**: The proposed memory compression merges features purely based on Euclidean similarity. While efficient, it lacks semantics (e.g., object-aware grouping or scene dynamics).
>
> **A3**:
> Thanks for this insightful comment. Our goal is to develop a simple, efficient, and effective memory compression strategy to main long-term information for ego-exo correspondence. For this purpose, we design the proposed memory compression method using Euclidean similarity. Despite its simplicity, it allows our model to effectively maintain useful long-term information for segmentation and execute efficiently in training and inference.
>
> However, we fully agree with the reviewer that incorporating higher-level cues, such as object-aware grouping, semantic relationships, or motion segmentation priors, could lead to a more informed and selective memory compression process. These enhancements have the potential to further improvement. We believe this is a highly promising direction for the ego-exo correspondence task, and leave exploring this semantic-aware memory compression design to our future work.
>
>  Again, thanks for this comment! We will add the above clarification in the revision.
>
> > **Q4**: **Modularity of MV-MoE**: The fusion module is tightly coupled with SAM2's architecture. It's not entirely clear how portable or reusable the module is with other segmentation backbones. This is not a big weakness, although it would be nicer is the module was such that it could be used with future versions of SAM2-like architectures.
>
> Tab. B: Application of MV-MoE on other segmentation backbones.
>
> | Model         | IoU ↑ | LE ↓  | CA ↑  | BA ↑  |
> |---------------|-------|-------|-------|-------|
> | STCN          | 27.39 | 0.109 | 0.378 | 61.97 |
> | STCN + MV-MoE | 30.51 | 0.083 | 0.436 | 62.81 |
> | QDMN          | 27.03 | 0.108 | 0.346 | 63.16 |
> | QDMN + MV-MoE | 28.75 | 0.100 | 0.376 | 64.02 |
>
> **A4**:
> Thanks for the insightful comment. While the proposed MV-MoE is initially designed on top of the SAM 2 architecture, its core component, i.e., the dual-branch routing mechanism that adaptively assigns contribution weights to each expert feature along both channel and spatial dimensions, is ***conceptually modular*** and can be ***readily integrated*** into other architectures that involve multi-source feature fusion. To demonstrate the generality of MV-MoE, we conduct additional experiments by integrating it into two alternative framework with different segmentation bones, including STCN and QDMN, for the Ego-Exo Correspondence task. The results on the Ego2Exo test split of the EgoExo4D dataset are shown in Tab. B.
>
> From Tab. B, we can clearly observe that, incorporating MV-MoE consistently enhances performance across metrics, indicating that the module is not only effective but also transferable to different backbone architectures. This suggests that MV-MoE can be easily adapted to other segmentation architectures/backbones and future versions of SAM 2-like frameworks.
>
> Once again, we sincerely appreciate the reviewer's valuable suggestion, which has helped improve the clarity and rigor of our work. We hope the additional experiments and clarifications satisfactorily address the reviewer's concerns.

---

### Official Review · Reviewer_HNsS · 2025-07-06

**Clarity:** 3
**Significance:** 3
**Originality:** 3
**Rating:** 4
**Confidence:** 3

**Summary:**

The paper tackles ego–exo object correspondence by extending Segment Anything Model 2 with a specialised framework called LM-EEC. The method introduces (i) a Memory-View Mixture-of-Experts module that dynamically fuses egocentric and exocentric features across channel and spatial dimensions, and (ii) a dual-memory bank with compression to preserve long-term context while pruning redundancy. Evaluated on EgoExo4D, LM-EEC sets new state-of-the-art performance and surpasses both SAM 2 and recent video-segmentation baselines

**Questions:**

Please check above weaknesses.

**Ethical Concerns:**

["NO or VERY MINOR ethics concerns only"]

**Final Justification:**

Most of the concerns I raised have been satisfactorily addressed during the rebuttal. After reviewing the feedback from other reviewers, I maintain a positive assessment of this paper.

**Limitations:**

Limitations have been provided in appendix.

**Paper Formatting Concerns:**

No paper formatting concerns.

**Quality:**

2

**Strengths And Weaknesses:**

**Strength**

- The manuscript is clear and engaging; the motivation naturally leads to the proposed methodology.

- Figures are well designed and effectively convey the pipeline, making the approach easy to follow.

- The Memory-View Mixture-of-Experts module is a thoughtful architectural addition that appears carefully engineered.

- The experimental section is thorough, combining strong baseline comparisons with detailed ablations that convincingly demonstrate the method’s effectiveness.

**Weaknesses**

- *Long-term correspondence evidence.* The core claim is improved long-term ego–exo correspondence, yet no explicit metric or qualitative example is provided. Please include quantitative results (e.g., association accuracy against clip length) or visualisations that illustrate successful long-range tracking.

- *Attribution of BA limitations.* The manuscript states that BA shortcomings “originate from existing baselines,” but the connection is not explained. Clarify precisely which baseline constraints carry over to LM-EEC and why they cannot be alleviated within your framework.

- *Inconsistent ablation scores.* Tables 2–6 are reported on the same validation set, yet the final scores differ. Explain whether these discrepancies arise from stochastic variation, different checkpoints, or another factor, and unify the reporting format to make cross-table comparisons straightforward.

---

> ### Author Rebuttal · Authors · 2025-07-31
>
> We sincerely thank the reviewer for the thoughtful and constructive feedback. We greatly appreciate the reviewer's recognition of the clarity of our manuscript, the soundness of our proposed framework, and the comprehensiveness of our experimental evaluations. Below, we provide detailed responses to each of the concerns the reviewer raised.
>
> > **Q1**: *Long-term correspondence evidence*. The core claim is improved long-term ego-exo correspondence, yet no explicit metric or qualitative example is provided. Please include quantitative results (e.g., association accuracy against clip length) or visualisations that illustrate successful long-range tracking.
>
> Tab. A: Association accuracy for our method and the baseline against different clip length.
>
> | Video Length   | < 200 frames | 200 - 600 frames | > 600 frames |
> |----------------|--------------|------------------|--------------|
> | Baseline Model | 0.5580       | 0.6260           | 0.4724       |
> | Our Model      | 0.6054       | 0.6732           | 0.6150       |
>
> **A1**:
> Thanks for this helpful comment. As suggested, to demonstrate the effectiveness of our long-term ego-exo correspondence, we conduct an experiment under the ego-exo correspondence setting in Tab. A to analyze the **association accuracy** of our method against different video clip length and compare it with the baseline (i.e., SAM 2) that excludes the core MV-MoE and the memory compression modules. In the experiment, the **association accuracy** is defined as the ratio of frames where the Intersection over Union (IoU) between the predicted and ground truth masks exceeds a threshold (set to 0.5). The comparison results are shown in Tab. A.
>
> As shown in Tab. A, our model consistently outperforms the baseline model across all video lengths. Notably, on the relatively longer videos with more than 600 frames, our model achieves a significant improvement of approximately 14% in association accuracy compared to the baseline model, which effectively validates the long-term ego-exo correspondence of our method.
>
> We appreciate the reviewer for this careful comment and will include the above analysis and results in the revision. Thanks again!
>
> > **Q2**: *Attribution of BA limitations*. The manuscript states that BA shortcomings "originate from existing baselines," but the connection is not explained. **(a)** Clarify precisely which baseline constraints carry over to LM-EEC and **(b)** why they cannot be alleviated within your framework.
>
> **A2**:
> Thanks for this insightful comment.
>
> **To sub-question (a):** The balanced accuracy (BA) primarily measures the ability of a model in determining if the object appears or not in the frame. In this paper, we adopt the architecture of SAM 2 as the baseline to develop our model. While SAM 2 (our baseline) is powerful in video segmentation, it **lacks** specific mechanisms to handle cases where the target object disappears from the scene. For example, it does not incorporate strategies to verify object presence or absence. As a consequence, our model inevitably inherits the limitation of SAM 2 to some extent. It is worth noting that, despite this, our method still achieves noticeable gains in BA compared to the baseline model, as shown in Tab. 1 of the paper.
>
> **To sub-question (b):** Thanks for this careful comment. In fact, this limitation can be alleviated within our framework (and also the baseline SAM 2 framework). A possible solution is to introduce a simple branch within our framework to predict the presence or absence of the object. However, since the primary focus of our work is on improving object-level correspondence and retaining informative long-term content. To this end, we introduce the memory-view mixture-of-experts (MV-MoE) module, which effectively fuses the memory-fused and view-guided information, as well as a dual-memory bank system with a dedicated compression strategy that preserves long-term information while removing redundancy. As demonstrated in our experiments, these components significantly improve the correspondence performance.
>
> We thank the reviewer and agree that handling object disappearance remains an important and challenging problem in ego-exo correspondence. We leave this to our future by exploring specific solution for BA. We will include the above analysis in the revision. Again, thanks!
>
> > **Q3**: *Inconsistent ablation scores*. Tables 2-6 are reported on the same validation set, yet the final scores differ. Explain whether these discrepancies arise from stochastic variation, different checkpoints, or another factor, and unify the reporting format to make cross-table comparisons straightforward.
>
> **A3**:
> Sorry for the confusion. The discrepancies in the final scores across Tab. 2-6 result from evaluations performed under *distinct controlled settings*. These settings are designed to isolate the effect of each individual module and eliminate potential interference from others, thereby allowing for a more intuitive validation of the motivation and effectiveness of each proposed component relative to the SAM 2 baseline. Below, we provide detailed clarifications for each table:
>
> **(i). Tab. 2**: To isolate the impact of the fusion mechanism (MV-MoE), we compare it using a basic FIFO strategy for memory storage upon the SAM 2 baseline, thereby excluding the influence of memory compression. The final score (highlighted in cyan) thus reflects the performance of the model with only the MV-MoE module.
>
> **(ii). Tab. 3 and 5**: After validating the individual and joint effectiveness of our proposed modules, we conduct further ablations on key hyperparameters, including the number of frames and memory size. The final scores (highlighted in cyan) in these tables correspond to our full model, integrating all proposed modules.
>
> **(iii). Tab. 4**: Here, we aim to analyze the memory storage strategy in isolation. To do so, we perform ablation on the SAM 2 baseline model, excluding both the MV-MoE and memory compression modules. This allows us to observe the individual contribution of memory storage without interference from other components.
>
> **(iv). Tab. 6**: To isolate the effect of the memory compression strategy, we adopt a simple addition fusion mechanism upon the SAM 2 baseline, removing the influence of MV-MoE. The final score (highlighted in cyan) represents the performance of the model with only the memory compression module.
>
>
> We thank the reviewer for this careful comment. As suggested, we will unify the reporting format for these ablation studies and clearly annotate which version of the model each score corresponds to, ensuring that cross-table comparisons are intuitive and consistent.
>
> Once again, we thank the reviewer for all the constructive feedback, which has significantly helped improve the clarity and rigor of our paper.

---

> > ### Comment · Reviewer_HNsS · 2025-08-06
> > **Rebuttal Comment by HNsS**
> >
> > Thank you for your thorough effort in addressing my concerns. I’m pleased to see that most issues have been resolved. I would like the points mentioned in the above rebuttal to be explained in detail in the final main script.

---

> > > ### Author Response · Authors · 2025-08-07
> > >
> > > Thanks for the reviewer’s feedback and for acknowledging our efforts in addressing the reviewer’s concerns. We are glad to hear that most issues have been resolved to the reviewer’s satisfaction. Rest assured, we will incorporate the detailed explanations of the points mentioned in the rebuttal into the final main script. Please let us know if there are any further clarifications or adjustments needed. Thanks again!

---

### Decision · Program_Chairs · 2025-09-17

**Decision:**

Accept (poster)

**Comment:**

The paper introduces LM-EEC for establishing object correspondence between egocentric and exocentric views in videos. It developed on top of SAM2 with two new technical designs: a Memory-View Mixture-of-Experts module and a dual-memory compression module. The designs are clearly presented, supported by strong empirical results on EgoExo4D, and validated through ablations. Reviewers consistently praised the clarity of writing, well-structured figures, and convincing baseline comparisons. All reviewers voted to accept the work. Congrats!

Concerns remain around generalization (evaluation is limited to a single dataset), efficiency of the memory compression strategy, and weaker performance in balanced accuracy for occluded/disappearing objects. Some of these are addressed through experiments during rebuttal. The main ones remained are the generalization of the method and improving BA. AC believes both are valid shortcomings of the work, and having both can potentially boost the impact of the work. However, the paper in its post-rebuttal format meets the bar for acceptance.